# Sulfonamide-directed site-selective functionalization of unactivated C(sp$^3$)−H enabled by photocatalytic sequential electron/proton transfer

Chaodong Wang [1], Zhi Chen[1], Jie Sun [1], Luwei Tong[1], Wenjian Wang[1], Shengjie Song[1] & Jianjun Li [1,2]✉

The generation of alkyl radical from C(sp$^3$)−H substrates via hydrogen atom abstraction represents a desirable yet underexplored strategy in alkylation reaction since involving common concerns remain adequately unaddressed, such as the harsh reaction conditions, limited substrate scope, and the employment of noble metal- or photo-catalysts and stoichiometric oxidants. Here, we utilize the synergistic strategy of photoredox and hydrogen atom transfer (HAT) catalysis to accomplish a general and practical functionalization of unactived C(sp$^3$)−H centers with broad reaction scope, high functional group compatibility, and operational simplicity. A combination of validation experiments and density functional theory reveals that the N-centered radicals, generated from free N − H bond in a stepwise electron/proton transfer event, are the key intermediates that enable an intramolecular 1,5-HAT or intermolecular HAT process for nucleophilic carbon-centered radicals formation to achieve heteroarylation, alkylation, amination, cyanation, azidation, trifluoromethylthiolation, halogenation and deuteration. The practical value of this protocol is further demonstrated by the gram-scale synthesis and the late-stage functionalization of natural products and drug derivatives.

The direct functionalization of unactivated C(sp$^3$)−H bonds has long been a focal point of chemical synthesis, allowing for the rapid construction of C(sp$^3$)−X (X = carbon or heteroatom) bonds in natural products and valuable drugs in a convenient and high atom-economic manner[1]. However, the intrinsic inertness of aliphatic C−H bonds as well as regioselectivity of multiple C−H bonds of similar chemical environments in feedstock alkanes, has posed formidable challenges for the development of highly demanding catalytic systems[2–4]. The past decades has witnessed tremendous development of inert aliphatic C−H bonds activation with the assistance of transition metals and directing groups, yet a high degree of unmet need remains[5–7]. Eminently, the

strategy that synergistically utilizing photoredox and HAT catalysis with electrophilic heteroatom-centered radicals (halogen, N, O, and S) has offered a complementary and potential avenue for the selective activation of inert aliphatic C−H bonds and subsequent functionalization[8–10], which was proverbially employed in well-established Hofmann-Löffler-Freytag (HLF) reaction[11]. Notwithstanding remarkable progress in this domain, the reactivity in this transformation is mainly restricted by the inherent difficulties in the formation of the heteroatom-centered radical species with directional effect, such as nitrogen-centered radicals (NCRs), which were generated typically from the prefunctionalized precursors of free N−H bond, such as N−halogen[12,13], N−N[14], N−O[15–17], and

[1]Key Laboratory for Green Pharmaceutical Technologies and Related Equipment of Ministry of Education, College of Pharmaceutical Sciences, Zhejiang University of Technology, Hangzhou, P. R. of China. [2]Taizhou Key Laboratory of Advanced Manufacturing Technology, Taizhou Institute, Zhejiang University of Technology, Taizhou, P. R. of China. ✉e-mail: lijianjun@zjut.edu.cn

**Fig. 1 | General strategies for NCRs-directed remote C(sp³)−H functionalization. a** N-directed remote C(sp³)−H activation from N-heteroatom precursors. **b** Remote C(sp³)−H activation from free amines based on PCET. **c** Our hypothesis: remote C(sp³)−H activation from free amines based on ET/PT. **d** Reaction design. **e** This work: remote C(sp³)−H functionalization through ET/PT. PG protecting group, FG functional group, EWG electron-withdrawing group.

N−S[18,19] precursors (Fig. 1a). In contrast, the generation of NCRs from free amines represents the most straightforward and desirable strategy but is thermodynamically challenging owing to the extraordinary stability of the N−H bonds (BDFEs > 100 kcal/mol)[20,21]. In this context, oxidative proton-coupled electron transfer (PCET) catalysis has gradually emerged as a reliable strategy for the general activation of N−H bonds[22–25], yet the application in HLF reaction is currently limited to the construction of remote carbon-carbon bonds (Fig. 1b)[26–32]. Thus, there still exists a clear impetus for developing a practical and robust photoredox catalytic platform that (1) can directly generate NCRs from non-prefunctionalized N−H bonds and (2) realize the diversified application of subsequent functionalization of unactivated aliphatic C−H bond in a site-selective manner.

In the relentless exploration of photocatalysis, a mechanistically distinct photocatalytic mode caught our attention, namely sequential electron/proton transfer (ET/PT), which can enable the activation of free N−H bonds to access high-energy NCRs. However, existing research based on this potent platform was mainly focused on the NCRs-engaged coupling or cascade cyclization to construct nitrogen-carbon bonds[33,34]. Based on the capability of NCRs to serve as the C(sp³) bond activators and the enormous potential of photoredox catalysis in the field of synthetic chemistry, we questioned whether an efficient, unactivated, selective C−H functionalization of aliphatic precursors could be achieved through photocatalytic ET/PT mode (Fig. 1c). On the other hand, the Minisci reaction involving the coupling between heteroarenes and nucleophilic alkyl radicals, is a powerful method to achieve the heteroarene diversification simply[35]. However, owing to the strong aliphatic C−H bonds and the net oxidative nature[36,37], previous studies on Minisci-type reaction between

heterocycles and C(sp³)−H donors were limited by the employment of prefunctionalized C(sp³)−H substrates, precious catalysts, and excessive amount of oxidant. Encouraged by the recent advancement in hydrogen-evolution cross coupling via the utilization of synergistic catalysis that combines photoredox-capable catalysts and transition metals[38–40], we believe that merging this synergistic catalysis manifold with the Minisci alkylation would afford an opportunity to overcome the above obstacles.

Herein, we disclose a photoredox-cobalt dual-catalyzed site-selective heteroarylation of unactive C(sp³)−H centers, in which N-centered sulfonamidyl radical intermediates generated through cleavage of N−H bonds in a photoinduced stepwise ET/PT process, are the key HAT catalysts for nucleophilic carbon-centered radical formation. A depiction of our reaction design appears in Fig. 1d. Given the extensive application of acridine-based photocatalysts in the functionalization of C(sp³)−H bonds due to their ability to produce active HAT radical species through photoinduced single electron oxidation[41–43], we thus envisioned that, in the presence of blue light (450-460 nm), the excited state of 9-mesityl-10-methylacridinium perchlorate (Mes-Acr⁺ClO₄⁻) ($E^*_{p/2}$ = +2.06 V vs SCE), is of oxidant enough to remove an electron from neutral amine substrate to afford the nitrogen radical cation **A**, which then further deproton to produce N-centered radical **B**. A commercially accessible cobaloxime catalyst [Co(dmgH)₂Py]Cl was introduced to recover the ground state of Mes-Acr⁺ from Mes-Acr•. The so-formed nitrogen radical could trigger intramolecular remote HAT through a cyclic transition state to afford the distal alkyl radical **C**. Subsequently, the addition of alkyl radical **C** to protonated hetero-arenes would give the alkylated intermediate **D**, which then remove a hydrogen atom in the catalysis of photochemically generated Co(II)

**Table 1 | Optimization of the reaction conditions[a]**

| Entry | Variation from the standard conditions | Yield (%)[b] |
|---|---|---|
| 1 | None | 62 |
| 2 | PC-1 (3 mol%) | 67 |
| 3 | Co(dmgH)₂PyCl (8 mol%) | 64 |
| 4 | PC-2 instead of PC-1 | 55 |
| 5 | PC-3, PC-4 or PC-5 instead of PC-1 | 14, trace, 23 |
| 6 | NaOAc, Cs₂CO₃ or K₃PO₄ instead of TFA | n.r. |
| 7 | ACN: HFIP = 3:1 | 86 |
| 8 | HFIP or DMSO instead of ACN | n.r. |
| 9 | DCM or DCE instead of ACN | 51, 56 |
| 10 | Without PC-1, [Co], TFA or light irradiation | n.r. |
| 11 | O₂ instead of Co(dmgH)₂PyCl | trace |
| 12 | PG² or PG⁹ instead of PG¹ | 75, 53 |
| 13 | PG³⁻⁵ or PG¹² instead of PG¹ | <5[c] |
| 14 | PG⁶⁻⁸, PG¹⁰ or PG¹¹ instead of PG¹ | n.r. |

*ACN* acetonitrile, *TFA* trifluoroacetic acid, *HFIP* 1,1,1,3,3,3-hexafluoro-2-propanol, *DMSO* dimethyl sulfoxide, *DCM* dichloromethane, *DCE* 1,2-dichloroethane, *n.r.* no reaction.
[a]Standard conditions: **1a** (0.2 mmol), **2a** (0.4 mmol), PC-1 (2 mol%), Co(dmgH)₂PyCl (5 mol%), TFA (0.4 mmol), ACN (2.0 mL, 0.1 M), 2 × 25 W blue LEDs, room temperature, under a nitrogen atmosphere, 24 h.
[b]Isolated yield.
[c]Yields were determined by analysis of the ¹H NMR spectra of the reaction mixture using 3,4,5-trichloropyridine as an internal standard.

species to deliver the target product **E** and hydrogen. In addition, the successful development in remote alkylation, amination, cyanation, azidation, trifluoromethylthiolation, halogenation, and deuteration of *N*-alkylsulfonamides based on ET/PT catalytic platform further emphasizes its versatility in synthetic chemistry (Fig. 1e), which is not only to realize the construction of useful structures but also expected to expedite the expansion of photoinduced ET/PT strategy in remote C(sp³)−H functionalization.

## Results
### Evaluation of the reaction conditions
To verify the feasibility of the vision, 2-phenylquinoline **1a** and 4-methoxy-*N*-pentylbenzenesulfonamide **2a** were chosen as model

substrates in the presence of 2.0 equivalents trifluoroacetic acid (TFA) and a catalytic amount of Mes-Acr⁺ClO₄⁻ (PC-1) and cobaloxime [Co(dmgH)₂Py]Cl in ACN at room temperature (Table 1). To our delight, the desired remote-coupled product **3** could be obtained in 62% isolated yield by reacting under the irradiation of blue light for 24 h (entry 1). Increasing the loading amount of photocatalyst and cobaloxime catalyst did not significantly improve the productivity of **3** (entries 2–3). Switching PC-1 with PC-2 resulted in slightly lower yields, whereas the excited reduction potential of PC-2 ($E^*_{p/2}$ = +2.20 V vs SCE) was more positive than PC-1 (entry 4). The yield of product **3** was dramatically diminished when employing PC-3, PC-4, or PC-5 as photocatalysts, which might be attributed to their low reduction potential so that they were unable to effectively engage a single electron

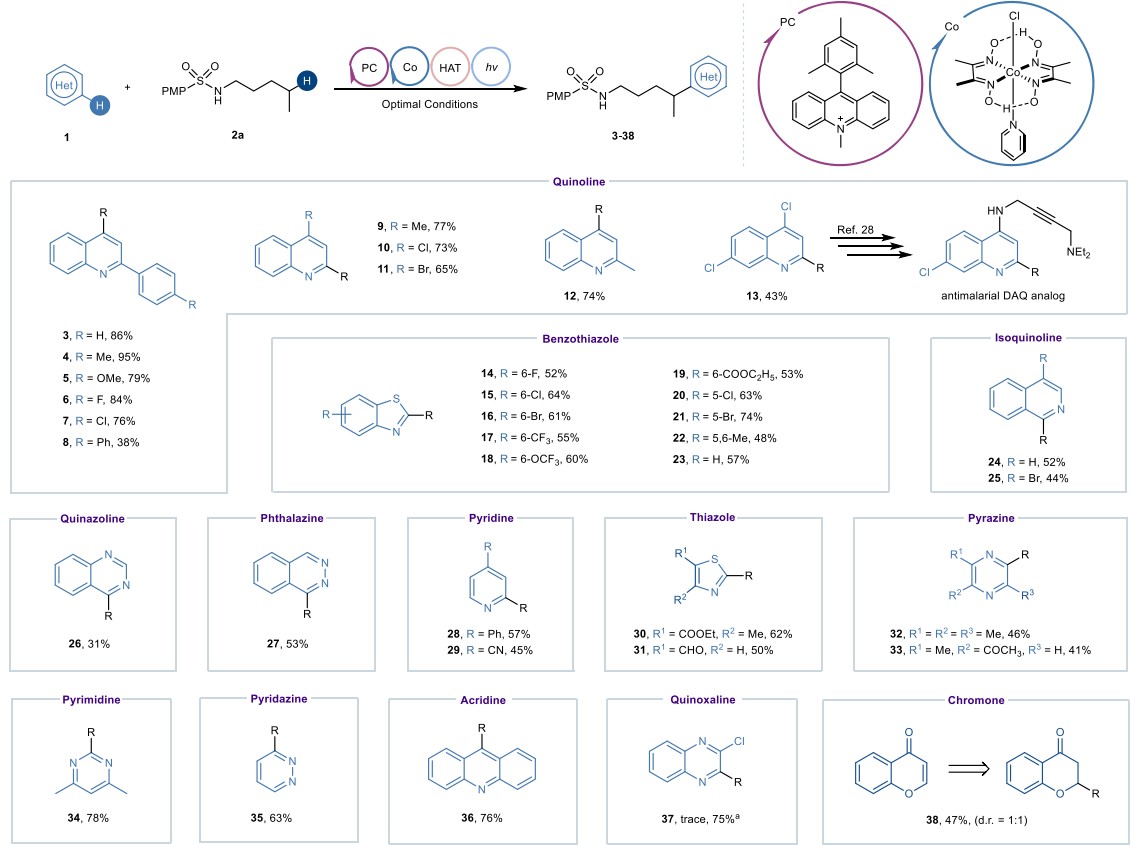

**Fig. 2 | Research on the scope of N-heteroarenes.** Reaction conditions: hetero-arenes **1** (0.2 mmol, 1.0 equiv), *N*-alkylsulfonamides **2a** (0.4 mmol, 2.0 equiv), Mes-Acr⁺ClO₄⁻ (0.004 mmol, 2 mol%), [Co(dmgH)₂Py]Cl (0.01 mmol, 5 mol%), TFA (0.4 mmol, 2.0 equiv), ACN/HFIP = 3:1 (2.0 mL, 0.1 M), 2 × 25 W blue LEDs ($\lambda$ = 450–460 nm), room temperature, under a N₂ atmosphere, 24 h. [a]K₂S₂O₈ (0.4 mmol, 2.0 equiv), TFA (0.4 mmol, 2.0 equiv), ACN/H₂O = 1:1 (2.0 mL, 0.1 M), 2 × 25 W purple LEDs ($\lambda$ = 390–400 nm), room temperature, under a N₂ atmosphere, 24 hours. PMP = 4-methoxybenzenesulfonyl.

oxidation with **2a** (entry 5). It was noteworthy that replacing the TFA by NaOAc, Cs₂CO₃, or K₃PO₄ completely inhibited this reaction, possibly due to the inertness of quinoline moiety under basic conditions (entry 6). Examination of a range of solvents showed that the reaction could be carried out efficiently in a mixed solvent of ACN and HFIP (entry 7), probably ascribed to the formation of hydrogen bond between N-centered radical and HFIP[44–46]. However, no product **3** was observed with HFIP or DMSO as sole solvent (entry 8). Other solvents such as DCE or DCM all gave **3** in lower yields (entry 9). Furthermore, the absence of Mes-Acr⁺ClO₄⁻, [Co(dmgH)₂Py]Cl, TFA, or blue light totally shut down the reaction (entry 10), demonstrating that each component plays a crucial role in promoting the reaction. Oxygen, as a green and abundant oxidant, has attracted our attention because it can not only act as an electron acceptor to promote the regeneration of photocatalysts[47], but also as a hydrogen acceptor in dehydrogenation coupling reactions[48]. We believed that replacing cobaloxime catalyst with oxygen would also achieve this goal. Unfortunately, only a trace amount of **3** was detected when the reaction was conducted under an oxygen atmosphere without the addition of [Co(dmgH)₂Py]Cl (entry 11). Finally, a survey of common N-protecting groups on amine substrates revealed 4-methoxybenzenesulfonyl as being optimal (entries 12-14).

## Substrate scope

Having established the viable catalyst system and conditions, we turned our attention to investigate the generality of N-heteroarenes **1** using 4-methoxy-*N*-pentylbenzenesulfonamide **2a** as a characteristic counterpart (Fig. 2). Quinolines substituted at the C2 or C4 positions reacted smoothly to give C4 or C2 coupling products **4-13** in fair to

high yields, respectively. The target product **13** can be readily trans-formed into 4-aminoquinoline, which could be served as an analog of active pharmaceutical ingredient for treating malarial[49]. Reactions of benzothiazole and its derivatives with various substituents proceeded smoothly with good regioselectivity to afford the desired products **14-23** in 48–74% yields regardless of their electronic properties and substitution patterns, showing good functional group tolerance in this cooperative catalysis. Some other medicinally important heterocycles, such as isoquinoline, quinazoline, and phthalazine, were all feasible substrates to give a variety of alkylated products **24-27** in moderate yields. It was noteworthy that monocyclic heteroarenes such as pyridine, thiazole, pyrazine, pyrimidine, and pyridazine were also compatible with this protocol, providing the monosubstituted coupling products **28-35** in 41-78% yields. Furthermore, fused polycyclic substrate acridine was successfully turned into the corresponding alkyl-coupled product **36** in good yield. Unexpectedly, 2-chloroquinoxaline failed to undergo this transformation, which might be due to its decomposition under the established reaction conditions. Gratify-ingly, the target product **37** was isolated in acceptable yield with K₂S₂O₈ as oxidant under irradiation of purple light for 24 hours. Interestingly, the reaction of chromone in such photocatalytic system would give the C2-alkyl substituted chromanone **38** in moderate yield.

We next explored the scope of N-alkylsulfonamides. As shown in Fig. 3a, a number of sulfonamides including both linear and cyclic amides were suitable substrates under the optimal conditions. Sulfo-namides carrying simple linear alkyl chains provided δ- and ε- sub-stituted regional isomers via 1,5-HAT and 1,6-HAT, wherein the elongation of the carbon chain was more conducive to the 1,5-HAT process, as demonstrated by the generation of products **39-42**.

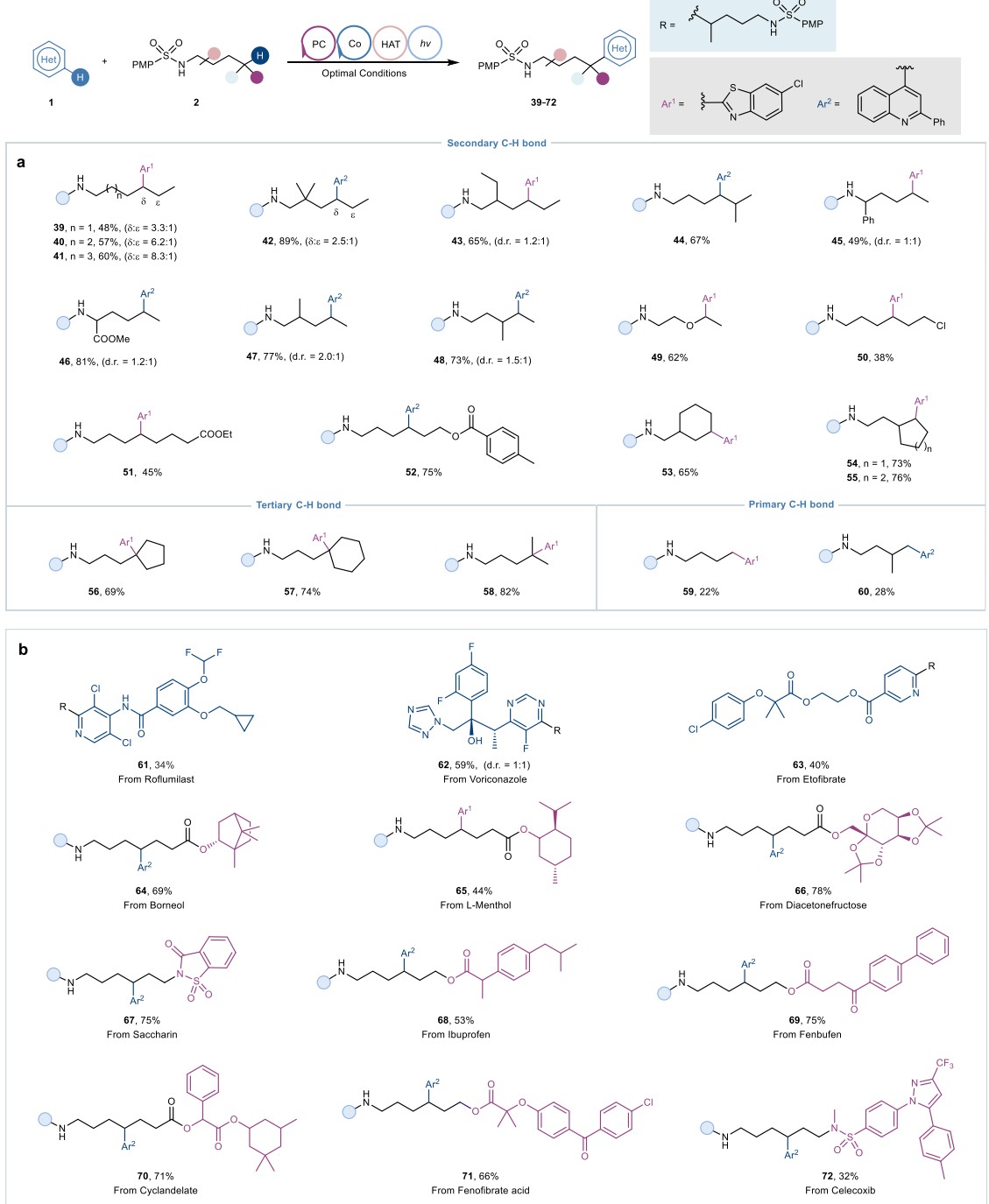

**Fig. 3 | Substrate scope for *N*-alkylsulfonamides, drug and natural product derivatives under optimal conditions. a** Scope of *N*-alkylsulfonamides. **b** Late-stage functionalization of drugs and complex compounds. Ar¹ 6-chlorobenzo[d]thiazole, Ar² 2-phenylquinoline.

Noticeably, products **43** and **44** were obtained with moderate yields in unique regioselectivity. Substrates substituted at the α-, β- and γ-positions of nitrogen proceeded smoothly and delivered good yields of the products **45-48**. Besides, functional groups, including ester, halogen, carbamate, and terminal benzoate, were proven to be tolerable, as demonstrated by the formation of products **49-52**. As expected, methylene C−H bonds of cyclic motifs could be successfully functionalized, affording the target products **53-55** in satisfactory yields. However, benzylic C−H bonds were not suitable under the current conditions, possibly due to the steric hindrance of phenyl moiety (please see Supplementary Fig. 2 for details). Moreover, tertiary

methine-containing and terminal methyl substrates were also viable abstraction partners, as demonstrated by the formation of products **56-60**, albeit the latter resulting in lower yield. Significantly, this method could be applied to the late-stage modification of complex natural products and drug derivatives (Fig. 4b). For instance, medicinally relevant heteroaromatic drugs, such as the core of Roflumilast (**61**), Voriconazole (**62**) and Etofibrate (**63**), could undergo alkylation modification efficiently. Derivatives of readily accessible natural products such as borneol, ʟ-menthol, Diacetonefructose, and saccharin behaved well to converted to the corresponding products (**64-67**) in moderate to high yields. Besides, a class of well-known nonsteroidal

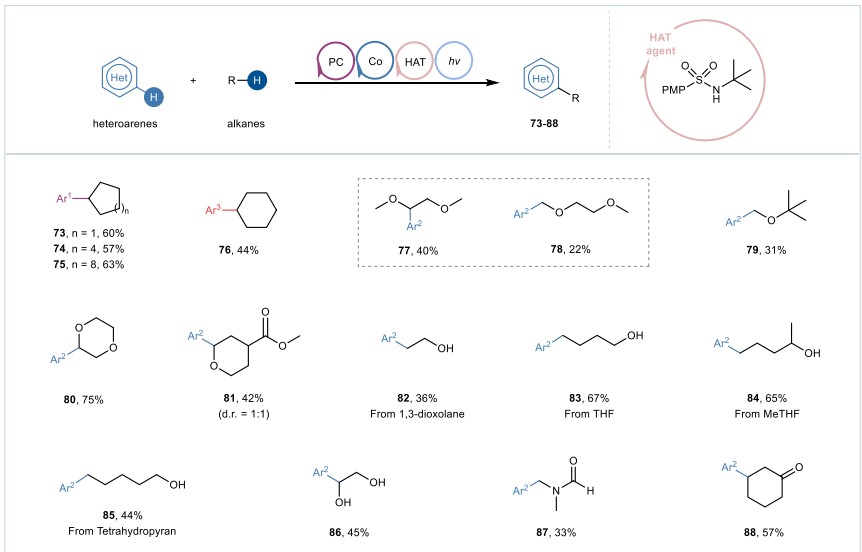

**Fig. 4 | NCR-mediated intermolecular Minisci alkylation.** Reaction conditions: heteroarenes **1** (0.2 mmol, 1.0 equiv), alkanes (0.2 mL), HAT agent (0.04 mmol, 20 mol%), Mes-Acr⁺ClO₄⁻ (0.004 mmol, 2 mol%), [Co(dmgH)₂Py]Cl (0.01 mmol, 5 mol%), TFA (0.4 mmol, 2.0 equiv), ACN/HFIP = 3:1 (2.0 mL), 2 × 25 W blue LEDs ($\lambda$ = 450−460 nm), room temperature, under a N₂ atmosphere, 24 hours. Ar¹ 6-chlorobenzo[d]thiazole, Ar² 2-phenylquinoline, Ar³ 4-methylquinoline.

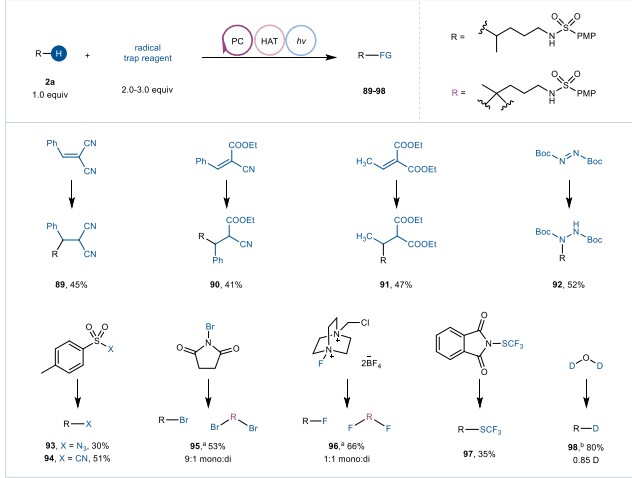

**Fig. 5 | Remote inert C(sp³)-H bonds functionalization.** Reaction conditions: **2a** (0.2 mmol, 1.0 equiv), radical trapping reagents (0.6 mmol, 3.0 equiv), Mes-Acr⁺ClO₄⁻ (0.006 mmol, 3 mol%), ACN/HFIP = 9:1 (2.0 mL, 0.1 M), 2 × 25 W blue LEDs ($\lambda$ = 450−460 nm), room temperature, under a N₂ atmosphere, 24 h. ᵃradical trapping reagents (0.4 mmol, 2.0 equiv), ᵇ1,2-diphenyldisulfane (0.04 mmol, 20 mol%), ACN/D₂O = 9:1 (2.0 mL, 0.1 M).

anti-inflammatory drugs, including Ibuprofen and Fenbufen, could provide the desired products **68** and **69** in good yields after protecting their carboxyl groups. Additionally, Cyclandelate, Fenofibrate, and Celecoxib were all compatible under the established conditions, giving the desired products **70**-**72** in appreciable yields.

In light of these results, we wonder if other kinds of C(sp³)–H substrates could be applied in such photo/cobalt dual catalyzed cross dehydrogenation coupling reaction via an intermolecular HAT process with *N*-alkylsulfonamide as an exogenous HAT catalyst (Fig. 4). To our delight, some simple cyclic alkanes such as cyclopentane, cyclohexane, cyclooctane and cyclododecane, could be heteroarylated smoothly to give the desired products **73**-**76** in moderate yields. Besides, the ethereal compounds **77**-**81** could be obtained under the optimized conditions. It was worth noting that ring-opening reactions could occur through C−O bond cleavage with cyclic ethers as coupling

partners, providing β-, δ- and ε-heteroarylated alcohols **82**-**85** in fair to good yields, which made this synergistic strategy more robust because these results were difficult to realize typically by dehydrogenation coupling between heteroarenes and free alcohols to the best of our knowledge. In addition, the functionalization of α-C(sp³)−H in amine and alcohol derivatives, and β-C(sp³)−H in ketone derivatives was also successful (products **86**-**88**).

To further demonstrate the versatility of this catalytic platform for the functionalization of remote inert C(sp³)−H bonds, we examined the reaction in the context of a range of radical trapping reagents under otherwise standard conditions (Fig. 5). Consistent with the results of classical PCET process, remote carbon radicals could still be produced smoothly and then engage in a conjugate addition reaction with an electron-deficient olefin partner to furnish a C(sp³)−C(sp³) bond under the catalysis of Mes-Acr⁺ClO₄⁻, as demonstrated by the formation of products **89**-**91**. Likewise, the so-formed carbon radical could undergo coupling with di-tert-butyl azodiformate (DBAD) to forge a C(sp³)−N bond (product **92**). Cyanides, azides and halides act as powerful synthons for organic chemistry due to their ability to feed into a variety of functional group interconversions. Therefore, the direct introduction of cyano, azide, and halogens into an unactive C(sp³)−H position is of great significance for drugs synthesis and modification. Delightfully, by using electron-deficient SOMO-philes, where a functional group is attached to an aryl sulfonyl group, we directly achieved inert C(sp³)−H bonds cyanation and azidation (products **93** and **94**). Additionally, the bromination and fluorination of inert C(sp³)−H bonds has been proven feasible by using *N*-bromosuccinimide and Selectfluor as coupling partners, respectively, as a mixture of mono- and diha-logenated products in the ratios of 9:1 and 1:1 (products **95** and **96**). Moreover, the strategy was further applied to the remote C(sp³)−H trifluoromethylthiolation in the presence of 2-((trifluoromethyl)thio) isoindoline-1,3-dione (product **97**). While the group of Studer and Xie independently reported the deuteration of unactivated C(sp³)−H bonds with amides as N-centered radical precursors[19,50], N-centered sulfonamidyl radical-triggered deuteration of unactivated C(sp³)−H bonds has not been reported to date. By introducing 1,2-diphe-nyldisulfane as a synergistic catalyst and D₂O as deuterium source, we were able to perform precise monodeuteration of *N*-alkylsulfo-namide in a yield of 80% (product **98**).

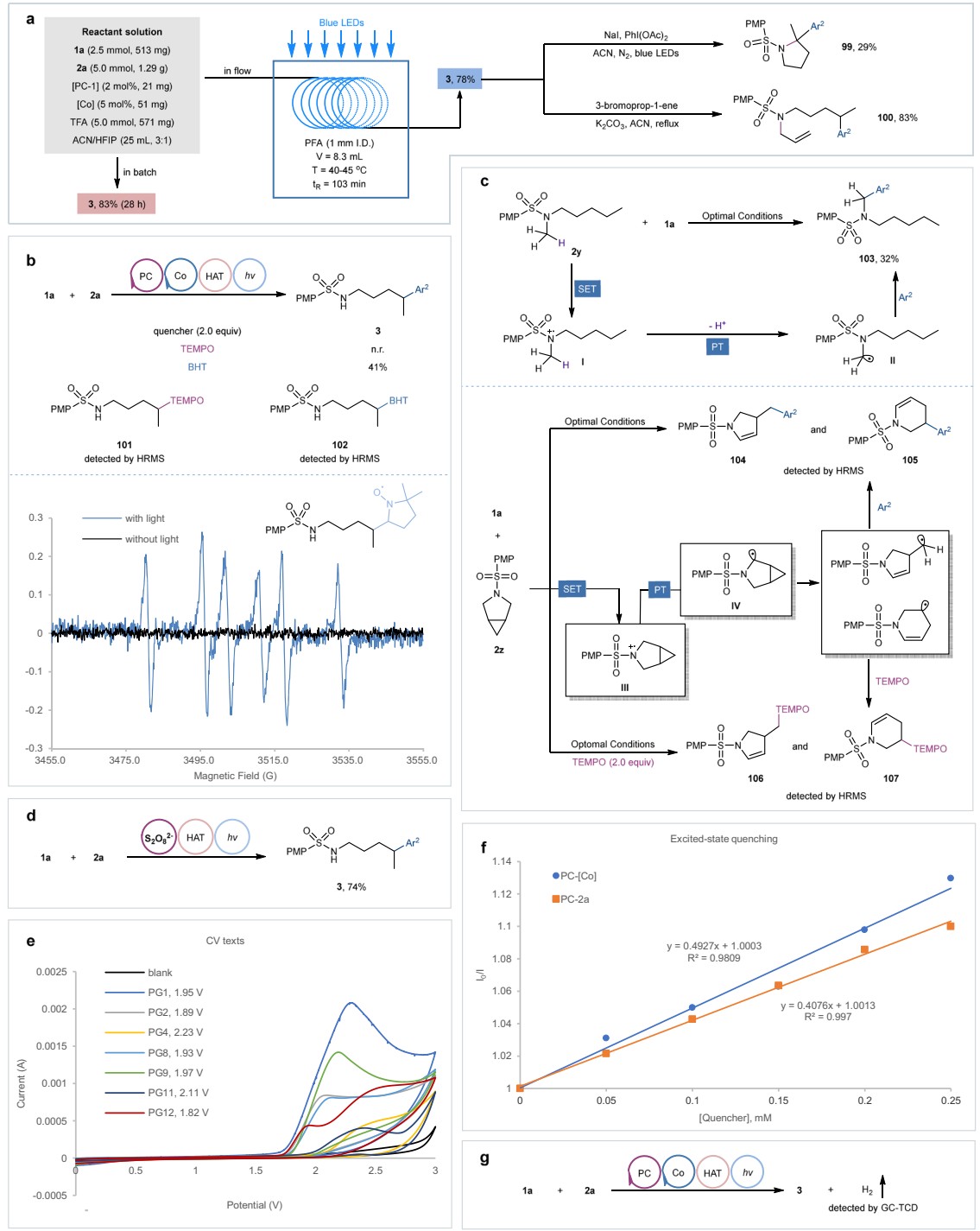

**Fig. 6 | Synthetic application and mechanistic studies.** See Supplementary Information for more detailed reaction conditions and descriptions, including: **a** Gram-scale experiments and product transformations. **b** Radical quenching experiments and EPR texts: spin-trapping experiments with DMPO. **c** Control and radical clock experiments. **d** K₂S₂O₈-promoted remote heteroarylation. **e** Cyclovoltammetric experiments. **f** Fluorescence quenching studies. **g** Hydrogen evolution detection. Ar² 2-phenylquinoline, TEMPO 2,2,6,6-tetramethylpiperidine 1-oxyl, BHT 3,5-di-*tert*-4-butylhydroxytoluene, DMPO 5,5-dimethyl-1-pyrroline *N*-oxide.

## Gram-scale synthesis and application of product

The scalability of the protocol was demonstrated by the gram-scale reaction performed on both in batch and continuous-flow conditions without significant erosion of the yield. The synthetic application of this protocol was further demonstrated by the versatile transformations of the resultant product **3**. For instance, treatment with NaI/PhI(OAc)₂ under light irradiation could smoothly convert the resultant product **3** into a pyrrolidine derivative **99** via the iodine-mediated

oxidative HLF cyclization. In addition, the N−H bond of **3** was easily proceeding a nucleophilic substitution with 3-bromoprop-1-ene under basic reaction conditions to produce a masked compound **100** in a high yield of 83% (Fig. 6a).

## Mechanistic investigations

A series of validation experiments were conducted to explore the reaction mechanism. We found that the reaction was significantly

suppressed in the presence of 2,2,6,6-tetramethylpiperidine 1-oxyl (TEMPO) and 3,5-di-tert-4-butylhydroxytoluene (BHT), and the δ-alkyl radical adducts **101** and **102** were respectively detected by HRMS, indicating the involvement of the radical species in this reaction. The formation of δ-alkyl radical was further confirmed by hyperfine structure spectrum analysis under EPR texts with 5,5-dimethyl-1-pyr-roline N-oxide (DMPO) as a radical capture agent. Noticeably, the radical signal was only observed under light irradiation, emphasizing the indispensable role of light in triggering the reaction (Fig. 6b). The methylated substrate **2y** failed to proceed with remote heteroaryla-tion, supporting the hypothesis that HAT process was enabled by NCRs generated by N−H bond cleavage. In contrast, heteroarylation occur-red at the α-position of nitrogen in methylated substrate **2y**, delivering the amidoalkylation product **103** in a yield of 32%, which implied that this transformation might involve the generation of less hindered α-aminoalkyl radical via the oxidation/deprotonation of tertiary amines[51,52]. With this insight, the radical clock experiments were carried out both under optimal conditions and in the case of TEMPO. Accordingly, the resultant products **104** and **105** were successfully detected by HRMS via α-aminoalkyl radical formation followed by radical-triggered ring-opening of cyclopropanes. Meanwhile, both the resultant products (**104** and **105**) and the radical adducts (**106** and **107**) were detected by HRMS in the case of TEMPO (Fig. 6c). All these results supported the presence of α-aminoalkyl radical in the control and radical clock experiments. Based on the previous reports[53], we assigned it to a sequential electrochemical-chemical event. Specifi-cally, neutral amines **2y** and **2z** underwent single electron oxidation to form sulfonamidyl radical cations **I** and **III**, which triggered deproto-nation occurred at the α-position to give C-centered radicals **II** and **IV** since the enhanced acidity of $C(sp^3)$−H bonds adjacent to the nitrogen atom. These enlightened results enabled us to assure that sulfonamidyl radical cations have been formed via single electron transfer between N-alkylsulfonamides bearing free N−H bonds and excited photo-catalyst, which then underwent N−H bonds cleavage to generate NCRs because the N−H bonds were more acidic than α-C−H bonds in such NCRs-triggered remote functionalization. However, the possibility that the sulfonamidyl radical cation directly mediates the intramolecular HAT cannot be ruled out[54]. To probe more details of the formation of NCRs, we performed the model reaction in the presence of $K_2S_2O_8$, which is an efficient oxidant widely applied in Minisci alkylation (Fig. 6d)[55–58]. Pleasingly, the remote heteroarylation proceeds effi-ciently, presenting the product **3** in a yield of 74%. Indeed, the gen-eration of NCR in this transformation involved a sequential ET/PT event as well, in which the single electron oxidation between $SO_4^{·-}$ ($E_{p/2} = +2.5–3.0$ V)[59] and 4-methoxy-N-pentylbenzenesulfonamide **2a** to produce sulfonamidyl radical cation was the key to trigger reaction (please see Supplementary Fig. 13 for more detailed descriptions about reaction mechanism)[60]. Cyclic voltammetry studies were next per-formed to provide more evidence that the generation of sulfonamidyl radical cations through single electron oxidation was thermo-dynamically feasible in our reaction system (Fig. 6e). In this case, the oxidation half-peak potential of 4-methoxy-N-pentylbenzenesulfona-mide (**2a**, PG[1]) was observed at +1.95 V (vs SCE in ACN), but upon the alteration of N-protecting groups on amine substrates, the half-peak potential of PG[4] (+2.23 V vs SCE in ACN) and PG[11] (+2.11 V vs SCE in ACN) was increased, and no redox features were displayed between 0 and 3.0 V (PG[3], PG[5], PG[6], PG[7], and PG[10]), which indicated that only 4-methoxy-N-pentylbenzenesulfonamide **2a** could productively undergo the single-electron transfer with excited acridine photo-catalyst. Despite the single electron transfer being thermodynamically permissible, the reactivity of PG[2] (1.89 V vs SCE in ACN) and PG[9] (1.97 V vs SCE in ACN) was inferior to PG[1]. It was worth noting that the remote heteroarylation of PG[8] (+1.93 V vs SCE in ACN) and PG[12] (+1.82 V vs SCE in ACN) did not proceed as effectively as PG[1], albeit with much lower oxidation half-peak potential. We attributed the latter to the much

higher bond dissociation energy of amidyl N−H bond, so that the production of corresponding NCR through N−H bond cleavage was energetically unfavorable[61,62]. Regretfully, we cannot provide a rea-sonable explanation for the former situation. Furthermore, Stern-Volmer quenching experiments were carried out by varying con-centrations of [Co(dmgH)₂Py]Cl, 2-phenylquinoline **1a**, and 4-meth-oxy-N-pentylbenzenesulfonamide **2a** in the presence of the acridine photocatalyst (Fig. 6f). It was found that the excited photocatalyst was not quenched by **1a**. On the other hand, both [Co(dmgH)₂Py]Cl and **2a** could quench the fluorescence of photo-excited Mes-Acr⁺ClO₄⁻, respectively. The quenching rates being directly proportional to their concentrations, which indicated the existence of single electron transfer between excited-state photocatalyst and [Co(dmgH)₂Py]Cl or **2a**. Considering the concentration of **2a** is much higher than Co cata-lyst, this reaction is preferentially initiated by the generation of reductive state of photocatalyst from the excited state of photo-catalyst via reductive quenching with N-alkylsulfonamides, whereas the Stern−Volmer quenching constant of [Co(dmgH)₂Py]Cl was slightly greater than that of the **2a**. Moreover, the hydrogen evolution was detected by GC-TCD analysis (Fig. 6g, please see Supplementary Fig. 31 for more details). Lastly, the light on-and-off experiment showed that continuous irradiation was essential for the product for-mation (please see Supplementary Fig. 32 for details).

## DFT studies

To better understand and validate this mechanistic hypothesis, density functional theory (DFT) calculations using the PBE0 hybrid functional[63] were performed in an investigation of the energetics of the proposed mechanism (Fig. 7). First, the 4-methoxy-N-pentyl-benzenesulfonamide **1a** is oxidized by the excited photocatalyst (PC⁺*) to form a sulfonamidyl radical cation **Int1** via a SET, accom-panied by the release of energy (4.6 kcal/mol). Subsequently, **Int1** undergoes deprotonation to afford the sulfonamidyl radical **Int2**. This step is exergonic by 13.9 kcal/mol. In this stage, one molecular HFIP can form two strong hydrogen bonds with one N-centered radical **Int2** to afford hydrogen-bonding complex **Int3**. Meanwhile, Co³ quenches the reduced photocatalyst (PC·) back to its ground state (PC⁺) with the release of energy (28.7 kcal/mol). These results suggest that both the stepwise ET/PT pathway and photocatalytic cycle are thermodynamically feasible. Later, a 1,5-HAT event pro-ceeds via **TS1**, generating a C-centered radical **Int4**. This step is exergonic by 19.6 kcal/mol and has an 8.9 kcal/mol energy barrier. The radical intermediate **Int4** then can attack C4 position of **1a-H⁺** to activate $C(sp^2)$−H bond and giving the additional intermediate **Int5**, which is then rearomatized by Co² species through a barrier of 18.5 kcal/mol to afford the **Int6** and **Int7**. The H₂ evolution between **Int6** and **Int7** requires it to overcome a 23.6 kcal/mol energy barrier, and liberates the cobaloxime catalyst and product **3**, which is iden-tified as the rate-determining step. Noticeably, the process that **Int7** accepts a proton provided by TFA to release H₂ seems more favor-able, proceeded with 19.3 kcal/mol energy barrier. The entire process of remote heteroarylation is exergonic by 12.9 kcal/mol. To reveal more details about the Co²-mediated rearomatisation of **Int5**, we further analyzed the spin density evolution during this step. It was found that the concerted **TS3** shows a partial reduction of the spin density of Co² (0.60) (i.e., a partial cobalt oxidation from II to III), and reduction of the highly delocalized spin density in the **Int5** to finally recover the aromaticity in the product, supporting that the rear-omatisation was conducted by Co² species through hydrogen atom extraction from **Int5** in an open-shell singlet transition state. Taken together, the bulk of these evidence supports the surmised stepwise ET/PT reaction pathways in remote $C(sp^3)$−H functionalization. Finally, as a logical extension based on the current mechanistic fra-mework, we believe that ET/PT mode will further enrich the devel-opment of unactive $C(sp^3)$−H functionalization through the rational

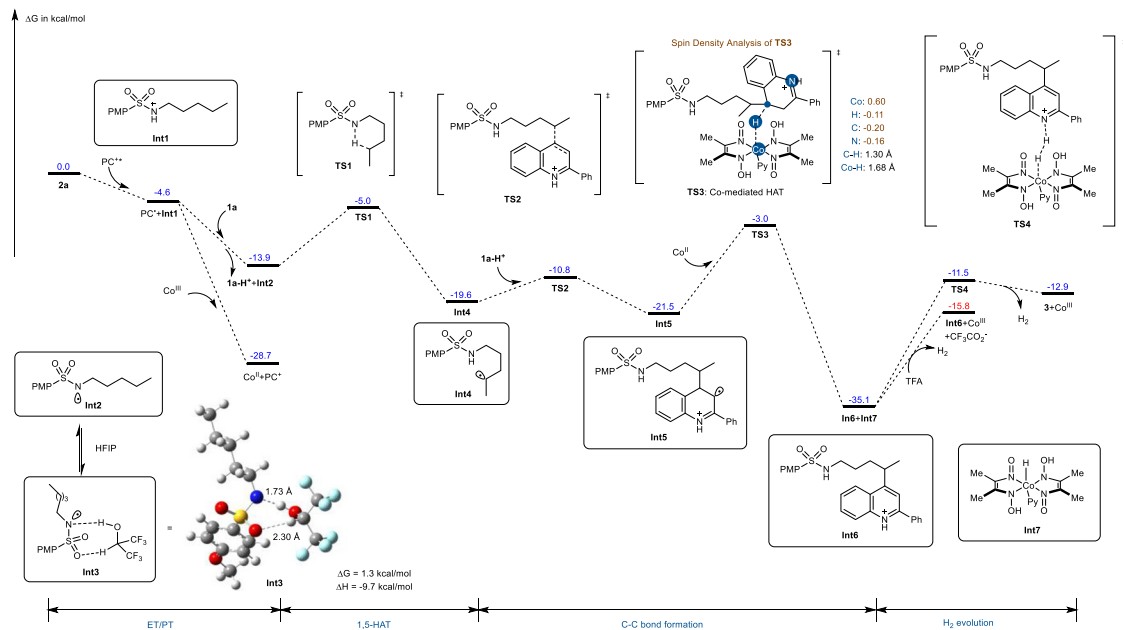

**Fig. 7 | Computational investigations.** Density functional theory calculations were performed at the PBE0-D3BJ/def2-TZVP + SMD(MeCN) level of theory. Energies are given in kcal/mol. Distances between the critical atoms are given in Å.

introduction of a radical trapping reagent or a cooperative catalyst that can match the oxidation potential of reductive state of acridine photocatalyst.

## Discussion

In summary, we have developed a photoinduced stepwise ET/PT pathway for N-centered sulfonamidyl radical generation via N−H bond cleavage. And based on this protocol, we have achieved heteroarylation, alkylation, amination, cyanation, azidation, trifluoromethylthiolation, halogenation, and deuteration of unactivated C(sp³)−H bonds through NCRs-triggered 1,5-HAT. Performed under mild and redox-neutral conditions, this protocol is atom- and step-economical and obviates the need of noble metal catalysts and photocatalysts. The tolerance for diverse functional groups, as well as natural products and drug fragments, could make this approach attractive for complex molecule modification or late-stage functionalization. Further development of new photochemical remote functionalization is still under way in our lab. Moreover, a series of validation experiments and DFT calculations provide strong support for the proposed mechanism. Given the ubiquity and relevance of *N*-alkylsulfonamides in synthetic and medicinal chemistry, we believed that this photoredox-catalyzed unactivated C(sp³)−H bonds functionalization mechanism would be of conceptual and practical interest to chemists in both academic and industrial settings.

## Methods
### General procedure for remote C(sp3)−H heteroarylation of sulfonamides

To a 10 mL Schlenk tube equipped with a magnetic stirring bar was added heteroarene **1** (0.2 mmol), N-protected amines substrates **2** (0.4 mmol), Acr⁺-Mes-ClO₄⁻ (2 mol%) and Co(dmgH)₂PyCl (5 mol%). After three cycles of evacuation and backfilling of the reaction flask with nitrogen, TFA (2.0 equiv.), ACN (1.5 mL) and HFIP (0.5 mL) were added to the tube under nitrogen. The mixture was then irradiated by two 25 W blue lamps for 24 h. The reaction mixture was quenched by adding 4 mL saturated NaHCO₃ solution and 15 mL water and then extracted with ethyl acetate (3 × 20 mL). The combined organic extracts were washed by brine, dried over Na₂SO₄, filtered,

concentrated under reduced pressure. The crude product was purified by column chromatography on silica gel to afford the desired product **3-72**.

### General procedure for intermolecular dehydrogenative C(sp³)-H heteroarylation of alkanes

To a 10 mL Schlenk tube equipped with a magnetic stirring bar was added heteroarene **1** (0.2 mmol), Acr⁺-Mes-ClO₄⁻ (2 mol%), Co(dmgH)₂PyCl (5 mol%) and *N*-(tert-butyl)−4-methoxybenzenesulfonamide (20 mol%). After three cycles of evacuation and backfilling of the reaction flask with nitrogen, TFA (2.0 equiv.), alkanes (0.2 mL), ACN (1.5 mL), and HFIP (0.5 mL) were added to the tube under nitrogen. The mixture was then irradiated by two 25 W blue lamps for 24 h. The reaction mixture was quenched by adding 4 mL saturated NaHCO₃ solution and 15 mL water and then extracted with ethyl acetate (3 × 20 mL). The combined organic extracts were washed with brine, dried over Na₂SO₄, filtered, and concentrated under reduced pressure. The crude product was purified by column chromatography on silica gel to afford the desired product **73-88**.

### General procedure for remote C(sp³)−H functionalization of sulfonamides

To a 10 mL Schlenk tube equipped with a magnetic stirring bar was added **2a** (0.2 mmol, 1.0 equiv.), radical trap reagent (0.4-0.6 mmol, 2.0-3.0 equiv.) and Acr⁺-Mes-ClO₄⁻ (3 mol%). After three cycles of evacuation and backfilling of the reaction flask with nitrogen, ACN (1.8 mL) and HFIP (0.2 mL) were added to the tube under nitrogen. The mixture was then irradiated by two 25 W blue lamps for 24 h. The reaction mixture was quenched by adding 15 mL water and then extracted with ethyl acetate (3 × 20 mL). The combined organic extracts were washed by brine, dried over Na₂SO₄, filtered, and concentrated under reduced pressure. The crude product was purified by column chromatography on silica gel to afford the desired product **89-97**.

## Data availability

All data are available from the corresponding author by request. The data generated or analyzed during the present study are included in this article and its Supplementary Information, including detailed

information on experimental procedures, mechanistic studies, DFT calculations, compound characterization data and NMR spectra. The raw data of EPR tests, CV tests, Stern-Volmer fluorescence quenching tests and Cartesian coordinates in DFT calculations are available from the Source Data. Source data are provided with this paper.

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

## Acknowledgements

We are grateful for the financial support from the National Natural Science Foundation of China (no. 22178314, J.L. and 21776254, J.L.). Furthermore, we thank Dr. Y. Weng for helpful discussions and Z. Liu from Shiyanjia Lab (www.shiyanjia.com) for the DFT calculations and GC-TCD analysis.

## Author contributions

J.L. designed and directed the project. C.W. supervised the work and wrote the manuscript. Z.C., J.S., L.T., W.W., and S.S. performed the experiments and mechanistic studies. All authors contributed to the analysis and interpretation of the data. C.W. and J.L. made manuscript revisions.

## Competing interests

The authors declare no competing interests.
