## [Peer Review File · Nature Communications]

Sulfonamide-Directed Site-Selective Functionalization of Unactivated C(sp³)-H Enabled by Photocatalytic Sequential Electron/Proton TransferReviewers' Comments:

Reviewer #1:

Remarks to the Author:

The present work describes a photochemical method for the functionalization of amines via the formation of nitrogen centered radicals (NCRs), followed by 1,5-HAT and Minisci-type addition to heteroarenes. Despite a very extensive and detailed reaction scope, the work lacks the conceptual novelty to be published in Nature Communications. There are very similar reports in the literature using photocatalysis or strong oxidants to functionalize the same starting materials through the same reaction pathway (NCR followed by HAT and Minisci: Chem. Sci., 2019, 10, 6915 (10.1039/c9sc02564b); J. Org. Chem. 2019, 84, 15777–15787 (10.1021/acs.joc.9b02502)), prior precedents which have not been cited in the manuscript. That being set, and even though the transformation will likely attract interest at the community it lacks the conceptual novelty required for publication into Nature Communications.

Reviewer #2:

Remarks to the Author:

This manuscript presents a strategy to functionalize unactivated C(sp³)-H bonds through a synergistic photoredox and hydrogen atom transfer (HAT) catalysis. The reaction demonstrated a broad substrate scope and high functional group compatibility. The study introduces an efficient method for generating nitrogen-centered radicals (NCRs) from free N-H bonds via a photoinduced stepwise electron/proton transfer (ET/PT) pathway, overcoming the common challenges associated with noble metal catalysts, stoichiometric oxidants, and limited substrate scope in alkylation reactions. The authors' work extends the synthetic utility of NCRs as C(sp³)-H bond activators and highlights the potential of photoredox catalysis in the field of synthetic chemistry.

Novelty and Significance: The development of a photoinduced stepwise ET/PT pathway for generating NCRs from non-prefunctionalized N-H bonds is both novel and significant. This method provides a more straightforward and thermodynamically feasible approach to NCR generation, which is critical for the selective activation and functionalization of inert aliphatic C-H bonds. The broad substrate scope, including heteroarylation, alkylation, amination, cyanation, azidation, trifluoromethylthiolation, halogenation, and deuteration, showcases the method's versatility and potential impact on synthetic and medicinal chemistry. The authors also demonstrated the reaction is amenable to gram-scale synthesis.

Limitations and Substrate Scope: While the authors have demonstrated a broad reaction scope and high functional group compatibility, only a few examples with the HAT took place intermolecularly. If the authors could provide more examples of functionalization through intermolecular HAT, it would greatly enhance the significance of the work.

Proposed Reaction Mechanism and Mechanistic Studies: The detailed mechanistic investigations, including radical trapping experiments, EPR spectroscopy, and DFT calculations, provide a strong basis for the proposed reaction mechanism involving NCRs-triggered 1,5-HAT. However, the authors proposed the formation of H₂ gas, which is only based on the results of DFT calculations. The authors should try to support their hypothesis by experimentally detecting the H₂ gas formation using NMR or GC.

Recommendation for Publication: Considering the novelty, the broad substrate scope, and the potential impact of this work, I recommend this manuscript for publication in Nature Communications after addressing the aforementioned suggestions.

Reviewer #3:

Remarks to the Author:

Summary: The authors report on a remote C(sp³)-H Minisci reaction of alkyl, aryl sulfonamides. The key reactivity relies on oxidation of a sulfonamide by a highly oxidizing photoredox catalyst. Subsequent deprotonation of the sulfonamide radical cation generates a sulfonamidyl radical, which can then perform a 1,5-HAT. The resultant carbon-centered radical can then add to a heteroarene by way of a Minisci reaction. A Co catalyst is used as an oxidant or HAT abstractor to terminate the Minisci and turnover the photocatalyst. The reaction is therefore completely catalytic. The authors demonstrate a large heteroarene scope as well as a sulfonamide scope with different functional groups on the backbone of the alkyl chain of the sulfonamide. Late-stage functionalization of drug compounds is also established. Additionally, use of a catalytic tert-butyl sulfonamide is also used to allow for an intermolecular Minisci reaction with simple alkanes. Comprehensive mechanistic studies were also conducted, providing key insights such as confirmation of the carbon-centered radical via TEMPO trapping, ring opening reactions to suggest radical cation /radical species, stoichiometric oxidation conditions, CV data on various protecting groups of the sulfonamide, and Stern-Volmer experiments to show quenching of the excited state by the sulfonamide and Co catalyst. DFT calculations were also conducted.

Recommendation: I think the work is a great fit at Nature Communications. The synthetic work is comprehensive as is the mechanistic work. I am not convinced direct oxidation of the sulfonamide is advantageous over PCET....maybe it leads to more solvent compatibility? The advantage of PCET was that you didn't have to use extremely oxidizing photocatalysts or strong oxidants. But either way, I do think the authors have developed a solid method that hasn't been preceded, and is entirely catalytic.

Comments:

1. The title is too general. It could be applicable to a number of different papers. It should be changed to be specific to the functionality.
2. This paper by the Knowles lab needs to be referenced: J.Am.Chem.Soc.2018, 140, 741–747
3. The authors use sulfamidyl radical cation but is the spelling 'sulfonamidyl radical cation'?
4. Why have the authors ruled out the sulfonamidyl radical cation doing the intramolecular HAT? Similar to the active HLF mechanism. Why are they so sure the subsequent deprotonation occurs and their active HAT species is the sulfonamidyl radical vs the sulfonamidyl radical? I think this is really difficult to tease out so I don't expect the authors to have an answer but something to consider when they discuss their mechanism.
5. I agree with the authors the excited state of the photocatalyst is likely oxidizing the sulfonamide. even though the Co catalyst quenches, the Co is present in catalytic amounts so I do think the sulfonamide is oxidizing preferentially due to concentrations.

REVIEWER COMMENTS

Reviewer #1 (Remarks to the Author):

The present work describes a photochemical method for the functionalization of amines via the formation of nitrogen centered radicals (NCRs), followed by 1,5-HAT and Minisci-type addition to heteroarenes. Despite a very extensive and detailed reaction scope, the work lacks the conceptual novelty to be published in Nature Communications. There are very similar reports in the literature using photocatalysis or strong oxidants to functionalize the same starting materials through the same reaction pathway (NCR followed by HAT and Minisci: Chem. Sci., 2019, 10, 6915 (10.1039/c9sc02564b); J. Org. Chem. 2019, 84, 15777–15787 (10.1021/acs.joc.9b02502)), prior precedents which have not been cited in the manuscript. That being set, and even though the transformation will likely attract interest at the community it lacks the conceptual novelty required for publication into Nature Communications.

Reviewer #2 (Remarks to the Author):

This manuscript presents a strategy to functionalize unactivated C(sp³)-H bonds through a synergistic photoredox and hydrogen atom transfer (HAT) catalysis. The reaction demonstrated a broad substrate scope and high functional group compatibility. The study introduces an efficient method for generating nitrogen-centered radicals (NCRs) from free N-H bonds via a photoinduced stepwise electron/proton transfer (ET/PT) pathway, overcoming the common challenges associated with noble metal catalysts, stoichiometric oxidants, and limited substrate scope in alkylation reactions. The authors' work extends the synthetic utility of NCRs as C(sp³)-H bond activators and highlights the potential of photoredox catalysis in the field of synthetic chemistry.

Novelty and Significance: The development of a photoinduced stepwise ET/PT pathway for generating NCRs from non-prefunctionalized N-H bonds is both novel and significant. This method provides a more straightforward and thermodynamically feasible approach to NCR generation, which is critical for the selective activation and functionalization of inert aliphatic C-H bonds. The broad substrate scope, including heteroarylation, alkylation, amination, cyanation, azidation, trifluoromethylthiolation, halogenation, and deuteration, showcases the method's versatility and potential impact on synthetic and medicinal chemistry. The authors also demonstrated the reaction is amenable to gram-scale synthesis.

Limitations and Substrate Scope: While the authors have demonstrated a broad reaction scope and high functional group compatibility, only a few examples with the HAT took place intermolecularly. If the authors could provide more examples of functionalization through intermolecular HAT, it would greatly enhance the significance of the work.

Proposed Reaction Mechanism and Mechanistic Studies: The detailed mechanistic

investigations, including radical trapping experiments, EPR spectroscopy, and DFT calculations, provide a strong basis for the proposed reaction mechanism involving NCRs-triggered 1,5-HAT. However, the authors proposed the formation of H₂ gas, which is only based on the results of DFT calculations. The authors should try to support their hypothesis by experimentally detecting the H₂ gas formation using NMR or GC.

Recommendation for Publication: Considering the novelty, the broad substrate scope, and the potential impact of this work, I recommend this manuscript for publication in Nature Communications after addressing the aforementioned suggestions.

Reviewer #3 (Remarks to the Author):

Summary: The authors report on a remote C(sp³)-H Minisci reaction of alkyl, aryl sulfonamides. The key reactivity relies on oxidation of a sulfonamide by a highly oxidizing photoredox catalyst. Subsequent deprotonation of the sulfonamide radical cation generates a sulfonamidyl radical, which can then perform a 1,5-HAT. The resultant carbon-centered radical can then add to a heteroarene by way of a Minisci reaction. A Co catalyst is used as an oxidant or HAT abstractor to terminate the Minisci and turnover the photocatalyst. The reaction is therefore completely catalytic. The authors demonstrate a large heteroarene scope as well as a sulfonamide scope with different functional groups on the backbone of the alkyl chain of the sulfonamide. Late-stage functionalization of drug compounds is also established. Additionally, use of a catalytic tert-butyl sulfonamide is also used to allow for an intermolecular Minisci reaction with simple alkanes. Comprehensive mechanistic studies were also conducted, providing key insights such as confirmation of the carbon-centered radical via TEMPO trapping, ring opening reactions to suggest radical cation/radical species, stoichiometric oxidation conditions, CV data on various protecting groups of the sulfonamide, and Stern-Volmer experiments to show quenching of the excited state by the sulfonamide and Co catalyst. DFT calculations were also conducted.

Recommendation: I think the work is a great fit at Nature Communications. The synthetic work is comprehensive as is the mechanistic work. I am not convinced direct oxidation of the sulfonamide is advantageous over PCET....maybe it leads to more solvent compatibility? The advantage of PCET was that you didn't have to use extremely oxidizing photocatalysts or strong oxidants. But either way, I do think the authors have developed a solid method that hasn't been precedented, and is entirely catalytic.

Comments:

1. The title is too general. It could be applicable to a number of different papers. It should be changed to be specific to the functionality.
2. This paper by the Knowles lab needs to be referenced: J.Am.Chem.Soc.2018, 140, 741-747
3. The authors use sulfamidyl radical cation but is the spelling 'sulfonamidyl radical cation'?
4. Why have the authors ruled out the sulfonamidyl radical cation doing the intramolecular HAT? Similar to the active HLF mechanism. Why are they so sure the subsequent

deprotonation occurs and their active HAT species is the sulfonamidyl radical vs the sulfonamidyl radical? I think this is really difficult to tease out so I don't expect the authors to have an answer but something to consider when they discuss their mechanism.

5. I agree with the authors the excited state of the photocatalyst is likely oxidizing the sulfonamide. even though the Co catalyst quenches, the Co is present in catalytic amounts so I do think the sulfonamide is oxidizing preferentially due to concentrations.

Point-by-Point Response to the Reviewers' Comments

We deeply acknowledge the three reviewers for their constructive and valuable comments they made to this work entitled "Site-Selective Functionalization of Unactivated C(sp³)-H Bonds via Synergistic Merger of Photoredox and HAT Catalysis" (Manuscript number: NCOMMS-24-04927-T). The point-by-point responses to the reviewer's comments are listed in *italic text in blue*.

Reviewer #1 (Remarks to the Author):

The present work describes a photochemical method for the functionalization of amines via the formation of nitrogen centered radicals (NCRs), followed by 1,5-HAT and Minisci-type addition to heteroarenes. Despite a very extensive and detailed reaction scope, the work lacks the conceptual novelty to be published in Nature Communications. There are very similar reports in the literature using photocatalysis or strong oxidants to functionalize the same starting materials through the same reaction pathway (NCR followed by HAT and Minisci: Chem. Sci., 2019, 10, 6915 (10.1039/c9sc02564b); J. Org. Chem. 2019, 84, 15777–15787 (10.1021/acs.joc.9b02502)), prior precedents which have not been cited in the manuscript. That being set, and even though the transformation will likely attract interest at the community it lacks the conceptual novelty required for publication into Nature Communications.

Author's reply:

We thank the reviewer for these comments, but we believe that our work has sufficient conceptual novelty. Many new developments in NCRs-triggered remote C(sp³)-H functionalization included in this manuscript are listed as follow:

- 1. By utilizing photoinduced ET/PT, we directly accomplished the NCRs formation through free N-H bonds cleavage without pre-functionalization and external base.*
- 2. By rational introducing a cooperative catalyst or a radical trapping reagent that can match the oxidation potential of reductive state of photocatalyst, we achieved inert C(sp³)-H heteroarylation, alkylation, amination, cyanation, azidation, trifluoromethylthiolation, halogenation and deuteration based on NCRs-triggered remote HAT.*
- 3. Performed under mild and redox-neutral conditions, this protocol is atom- and step-economical and obviates the need of noble metal catalysts and stoichiometric strong oxidants.*
- 4. In addition to establishing a simple and efficient catalytic system, the practical value of this protocol was further demonstrated by gram-scale synthesis conducted both in batch and flow and the late-stage functionalization of drugs and natural products.*
- 5. We have made a detailed and comprehensive proof experiment to explore the reaction mechanism.*

We do notice that, as the reviewer pointed out, the group of Zhu (Chem. Sci., 2019, 10, 6915-6919) and Chen (J. Org. Chem. 2019, 84, 15777–15787) have made a contribution to the remote heteroarylation of amines. However, these transformations were limited by

the employment of stoichiometric strong oxidants. With regard to the reaction mechanism, in Zhu's work, the key of high iodine reagent-involved NCRs generation lies in the photoinduced homolysis of N–I bonds formed in situ between free amines and PIFA; and in Chen's work, there is not enough evidence to show that the NCRs were formed via photocatalytic PCET or SET. In our work, we clearly described a PC/Co dual-catalyzed remote heteroarylation based on sequential electron/proton transfer mechanism. More importantly, this photoinduced ET/PT mode could be further developed in remote alkylation, amination, cyanation, azidation, trifluoromethylthiolation, halogenation and deuteration of N-alkylsulfonamides. To the best of our knowledge, this is the first case of NCRs-driven unactivated C(sp³)–H functionalization through synergistic utilization of photoinduced ET/PT and hydrogen atom transfer catalysis.

In summary, we believe without doubt that our work is conceptually novel. We sincerely hope that our work can get your approval. Meanwhile, the references (Refs. 13 and 32) mentioned by reviewer have been cited in the revised manuscript.

Reviewer #2 (Remarks to the Author):

This manuscript presents a strategy to functionalize unactivated C(sp³)–H bonds through a synergistic photoredox and hydrogen atom transfer (HAT) catalysis. The reaction demonstrated a broad substrate scope and high functional group compatibility. The study introduces an efficient method for generating nitrogen-centered radicals (NCRs) from free N–H bonds via a photoinduced stepwise electron/proton transfer (ET/PT) pathway, overcoming the common challenges associated with noble metal catalysts, stoichiometric oxidants, and limited substrate scope in alkylation reactions. The authors' work extends the synthetic utility of NCRs as C(sp³)–H bond activators and highlights the potential of photoredox catalysis in the field of synthetic chemistry.

Author's reply:

We thank the reviewer for this positive assessment of our work.

Novelty and Significance: The development of a photoinduced stepwise ET/PT pathway for generating NCRs from non-prefunctionalized N–H bonds is both novel and significant. This method provides a more straightforward and thermodynamically feasible approach to NCR generation, which is critical for the selective activation and functionalization of inert aliphatic C–H bonds. The broad substrate scope, including heteroarylation, alkylation, amination, cyanation, azidation, trifluoromethylthiolation, halogenation, and deuteration, showcases the method's versatility and potential impact on synthetic and medicinal chemistry. The authors also demonstrated the reaction is amenable to gram-scale synthesis.

Author's reply:

We thank the reviewer for this positive appraisal of our work.

Limitations and Substrate Scope: While the authors have demonstrated a broad reaction scope and high functional group compatibility, only a few examples with the HAT took place intermolecularly. If the authors could provide more examples of functionalization through intermolecular HAT, it would greatly enhance the significance of the work.

Author's reply:

We are grateful for this reviewer's constructive comment. We follow the reviewer's kind suggestion and have further tested the scope of diverse sp^3 C-H donors in NCR-mediated intermolecular Minisci alkylation. The results indicated that cyclopentane, 1,2-dimethoxyethane, 2-methoxy-2-methylpropane, methyl tetrahydro-2H-pyran-4-carboxylate, ethane-1,2-diol, *N,N*-dimethylformamide and cyclohexanone could be successfully converted into corresponding coupling products (products **73**, **77**, **78**, **79**, **81**, **86**, **87** and **88**) under the same conditions (please see Page 8, Fig.4 in the revised manuscript for details). The data characterization and spectra of relevant compounds have been supplemented in Supplementary Information (Page S111, S113-S117, S237, S241-S243, S245, S250-S252).

Fig.4 NCR-mediated intermolecular Minisci alkylation. Reaction conditions: heteroarenes **1** (0.2 mmol, 1.0 equiv), alkanes (0.2 mL), HAT agent (0.04 mmol, 20 mol%), Mes-Acr⁺ClO₄⁻ (0.004 mmol, 2 mol%), [Co(dmgh)₂Py]Cl (0.01 mmol, 5 mol%), TFA (0.4 mmol, 2.0 equiv), ACN/HFIP = 3:1 (2.0 mL, 0.1 M), 2 × 25 W blue LEDs (λ = 450–460 nm), room temperature, under a N₂ atmosphere, 24 hours. Ar¹ = 6-chlorobenzo[*d*]thiazole, Ar² = 2-phenylquinoline, Ar³ = 4-methylquinoline.

Proposed Reaction Mechanism and Mechanistic Studies: The detailed mechanistic investigations, including radical trapping experiments, EPR spectroscopy, and DFT calculations, provide a strong basis for the proposed reaction mechanism involving NCRs-triggered 1,5-HAT. However, the authors proposed the formation of H₂ gas, which is only based on the results of DFT calculations. The authors should try to support their hypothesis by experimentally detecting the H₂ gas formation using NMR or GC.

Author's reply:

We agree and appreciate the reviewer for this valuable comment. As per the reviewer's helpful suggestion, we have tested the H₂ gas formation using GC-TCD. The results indicated the formation of H₂ during the reaction (please see Page S30, Figure S31 in Supplementary Information for details).

Figure S31. Hydrogen evolution detected by GC-TCD

Recommendation for Publication: Considering the novelty, the broad substrate scope, and the potential impact of this work, I recommend this manuscript for publication in Nature Communications after addressing the aforementioned suggestions.

Author's reply:

We thank the reviewer for the kind and positive comments.

Reviewer #3 (Remarks to the Author):

Summary: The authors report on a remote C(sp³)-H Minisci reaction of alkyl, aryl sulfonamides. The key reactivity relies on oxidation of a sulfonamide by a highly oxidizing photoredox catalyst. Subsequent deprotonation of the sulfonamide radical cation generates a sulfonamidyl radical, which can then perform a 1,5-HAT. The resultant carbon-centered radical can then add to a heteroarene by way of a Minisci reaction. A Co catalyst is used as an oxidant or HAT abstractor to terminate the Minisci and turnover the photocatalyst. The reaction is therefore completely catalytic. The authors demonstrate a large heteroarene scope as well as a sulfonamide scope with different functional groups on the backbone of the alkyl chain of the sulfonamide. Late-stage functionalization of drug compounds is also established. Additionally, use of a catalytic tert-butyl sulfonamide is also used to allow for an intermolecular Minisci reaction with simple alkanes. Comprehensive

mechanistic studies were also conducted, providing key insights such as confirmation of the carbon-centered radical via TEMPO trapping, ring opening reactions to suggest radical cation/radical species, stoichiometric oxidation conditions, CV data on various protecting groups of the sulfonamide, and Stern-Volmer experiments to show quenching of the excited state by the sulfonamide and Co catalyst. DFT calculations were also conducted.

Recommendation: I think the work is a great fit at Nature Communications. The synthetic work is comprehensive as is the mechanistic work. I am not convinced direct oxidation of the sulfonamide is advantageous over PCET....maybe it leads to more solvent compatibility? The advantage of PCET was that you didn't have to use extremely oxidizing photocatalysts or strong oxidants. But either way, I do think the authors have developed a solid method that hasn't been preceded, and is entirely catalytic.

Author's reply:

We thank the reviewer for the kind, positive and insightful comments. We agree the view pointed by the reviewer that PCET strategy has unique advantages in the generation of NCRs. However, the requirement for Brønsted bases precludes the application of this strategy to the transformations that are incompatible under basic conditions, such as Minisci-type reaction. In our work, the direct oxidation of sulfonamides to form NCRs via ET/PT event avoid the employment of Brønsted bases and enable the diversity in subsequent C(sp³)-H bond functionalization. But either way, we do believe the reviewer's comment is of great guiding significance for our future research.

Comments:

1. The title is too general. It could be applicable to a number of different papers. It should be changed to be specific to the functionality.

Author's reply:

We thank the reviewer's constructive comment. We have changed the title of revised manuscript as "Sulfonamide-Directed Site-Selective Functionalization of Unactivated C(sp³)-H Enabled by Photocatalytic Sequential Electron/Proton Transfer".

2. This paper by the Knowles lab needs to be referenced: J. Am. Chem. Soc. 2018, 140, 741-747

Author's reply:

We appreciate the reviewer for this valuable comment. We follow the reviewer's helpful suggestion and have supplemented the aforementioned reference in the revised manuscript (Page 1, Ref. 25).

3. The authors use sulfamidyl radical cation but is the spelling 'sulfonamidyl radical cation'?

Author's reply:

We are grateful for the reviewer's careful reading. We have checked the full paper and corrected all the mistakes.

4. Why have the authors ruled out the sulfonamidyl radical cation doing the intramolecular HAT? Similar to the active HLF mechanism. Why are they so sure the subsequent

deprotonation occurs and their active HAT species is the sulfonamidyl radical vs the sulfonamidyl radical? I think this is really difficult to tease out so I don't expect the authors to have an answer but something to consider when they discuss their mechanism.

Author's reply:

We thank the reviewer's expert comment. On the one hand, we should indeed note the possibility that the sulfonamidyl radical cation could be served as HAT agent based on the precedent studies that aminium radical cation is sufficiently polarized to enable the abstraction of a hydrogen atom from the δ carbon, producing a new C-centered radical (Hofmann AW. Chem. Ber. 1883, 16, 558-560; J. Am. Chem. Soc. 1960, 82, 1657-1668; Chem. Rev. 1963, 63, 55-64), as shown in the below figure. On the other hand, there is no direct evidence that sulfonamide radical cation could be served as HAT reagent to activate C(sp³)-H bonds. Recently, the innovative breakthrough of Lei's group in in-situ EPR technology provides an effective means for the detection of N-centered sulfonamidyl radicals (J. Am. Chem. Soc. 2021, 143, 20863-20872). However, due to the limited experiment conditions, it's hard for us to provide the hyperfine structure of sulfonamidyl radical via in situ EPR test. Anyway, we cannot rule out the possibility that sulfonamidyl radical cation do the intramolecular HAT. We thus follow the reviewer's suggestion and have made the corresponding supplement in the section of Mechanistic investigations of the revised manuscript (Page 10).

Aminium radical cation-promoted remote hydrogen atom abstraction.

5. I agree with the authors the excited state of the photocatalyst is likely oxidizing the sulfonamide. even though the Co catalyst quenches, the Co is present in catalytic amounts so I do think the sulfonamide is oxidizing preferentially due to concentrations.

Author's reply:

We thank the reviewer for this insightful discussion. We agree the reviewer's view that the concentration difference between sulfonamide and Co catalyst is the key reason to explain that sulfonamide is preferentially oxidized by Co catalyst. We have changed the relevant description of the mechanism to "Considering the concentration of **2a** is much higher than Co catalyst, this reaction is preferentially initiated by the generation of reductive state of photocatalyst from excited state of photocatalyst via reductive quenching with N-alkylsulfonamides, whereas the Stern–Volmer quenching constant of $[\text{Co}(\text{dmgH})_2\text{Py}]\text{Cl}$

was slightly greater than that of the 2a” in the revised manuscript (Page 11).

We thank again for the reviewer’s valuable time and comments.

Reviewers' Comments:

Reviewer #2:

Remarks to the Author:

The authors have satisfactorily addressed all of my concerns raised during the previous review. The revisions have enhanced the clarity and robustness of the manuscript. Therefore, I recommend that the paper be published in its current form.

Reviewer #3:

Remarks to the Author:

The authors have kindly made the changes suggested by this reviewer, as well as the others. I hold my original opinion that the paper is a good fit for Nature Communications. As mentioned, the synthetic scope is comprehensive and thorough. It represents an advance in further exploration of sulfonamidyl radical chemistry.